# PROTECTING MINORITIES IN DIFFUSION MODELS VIA CAPACITY ALLOCATION

## ABSTRACT

Diffusion models have advanced quickly in image generation. However, their performance declines significantly on the imbalanced data commonly encountered in real-world scenarios. Current research on imbalanced diffusion models focuses on improving the objective function to facilitate knowledge transfer between majorities and minorities, thereby enhancing the generation of minority samples. In this paper, we make the first attempt to address the imbalanced data challenges in diffusion models from the perspective of model capacity. Specifically, majorities occupy most of the model capacity because of their larger representation, consequently restricting the capacity available for minority classes. To tackle this challenge, we propose Protecting Minorities via Capacity ALLocation (CALL). We reserve capacity for minority expertise by low-rank decomposing the model parameters and allocate the corresponding knowledge to the reserved model capacity through a capacity allocation loss function. Extensive experiments demonstrate that our method, which is orthogonal to existing methods, consistently and significantly improves the robustness of diffusion models on imbalanced data.

## 1 INTRODUCTION

In recent years, diffusion models have demonstrated exceptional potential and effectiveness in image generation, leading to increasing adoption by both industry and individuals (Ho et al., 2020; Song et al., 2021b; Dhariwal & Nichol, 2021). Diffusion model-based products such as DALL-E 2 (Ramesh et al., 2022) and the open-source Stable Diffusion (SD) (Rombach et al., 2022) have drawn millions of users, with numbers continuing to rise. However, recent studies reveal that diffusion models suffer from significant performance degradation when trained on class-imbalanced datasets (Qin et al., 2023; Zhang et al., 2024), which is particularly concerning given the prevalence of the imbalance nature in the real-world scenarios (Reed, 2001; Zhang et al., 2023).

Current research on imbalanced learning primarily focuses on improving the robustness of discriminative models (Buda et al., 2018; He & Garcia, 2009; Wang et al., 2021a; Menon et al., 2021; Cui et al., 2021) or generative adversarial networks (GANs) (Rangwani et al., 2021; 2022) to class imbalance. However, most of them cannot be directly applied to diffusion models due to the significantly different model structures and training and inference processes. For imbalanced diffusion models, existing efforts attempt to enhance the robustness to imbalanced distributions by improving the objective function. Class Balancing Diffusion Models (CBDM) (Qin et al., 2023) introduced a loss function regularizer that implicitly encourages generated images to follow a balanced prior distribution at each sampling step. Yan et al. (2024) designed a contrastive learning regularization to enhance inter-class separability. Oriented Calibration (OC) (Zhang et al., 2024) enhanced the generation quality of minorities through knowledge transfer between majorities and minorities.

In this paper, while existing efforts have primarily focused on the objective function, we approach the challenges of class-imbalanced diffusion models from a new perspective: *model capacity*. In scenarios with significant class imbalance, majority classes dominate most of the model capacity due to their larger representation, squeezing the capacity available for minority classes. As shown in Figure 1(a), minority classes experience a more pronounced change in loss before and after pruning the trained model. This behavior indicates that minority classes utilize less of the model's capacity, making them more vulnerable to pruning. We aim to enhance the robustness of diffusion models against imbalanced data by safeguarding the model capacity for minorities.

To address the challenge of model capacity encroachment, we propose a new method for imbalanced diffusion models: Protecting Minorities via Capacity ALLocation (CALL). Our core concept is to allocate dedicated model capacity for minority expertise, reserved in advance to prevent encroachment by majorities, thereby safeguarding the training process of minority samples. Specifically, we first decompose the model parameters into two parts using low-rank techniques: one for majority and general knowledge, and the other reserved for minority expertise (Eq. (3)). By introducing the capacity allocation loss (Eq. (4)), we effectively allocate the corresponding knowledge to the reserved model capacity during training. Due to the nature of low-rank parameter decomposition and aggregation, the capacity allocation does not introduce additional inference latency, which is crucial for real-world deployment of diffusion models. Additionally, CALL is orthogonal to existing methods and can be combined to achieve further improvements. The contributions are summarized as:

- We explore the challenge of imbalanced diffusion models from a new perspective: model capacity. We highlight that the key lies in protecting the model capacity allocated to minorities, setting it apart from existing efforts focusing on improving the objective function to enhance minorities.

- To tackle the issue of majorities encroaching on the model capacity required for minorities, propose a novel method, CALL, which protects minorities by reserving model capacity for minority expertise and effectively allocating the corresponding knowledge during training. CALL is orthogonal to existing methods, allowing for complementary integration.

- We conduct extensive experiments to showcase the superiority of our method, CALL, in enhancing the robustness of diffusion models against imbalanced data across various settings, including training diffusion models from scratch and fine-tuning pre-trained Stable Diffusion.

## 2 RELATED WORK

**Diffusion Models.** Diffusion models, a powerful class of generative models, are originally inspired by non-equilibrium thermodynamics (Sohl-Dickstein et al., 2015) and are now successfully applied to image generation (Ho et al., 2020), showing remarkably effective performance (Dhariwal & Nichol, 2021; Rombach et al., 2022). Ho et al. (2020) conduct the training of diffusion models using a weighted variational bound. (Song et al., 2021b) propose an alternative method for constructing diffusion models by using a stochastic differential equation (SDE). Karras et al. (2022) introduce a design space that clearly outlines the key design choices in previous works. Denoising diffusion implicit models (DDIMs) (Song et al., 2021a) employs an alternative non-Markovian generation process, enabling faster sampling for diffusion models.

**Imbalanced Generation.** Several works have investigated imbalanced generation based on generative adversarial networks (GANs) (Goodfellow et al., 2014). CB-GAN (Rangwani et al., 2021) mitigates class imbalance during training by using a pre-trained classifier. Rangwani et al. (2022) note that performance decline in long-tailed generation mainly results from class-specific mode collapse in minority classes, which is linked to the spectral explosion of the conditioning parameter matrix. To address this, they propose a corresponding group spectral regularizer. With diffusion models demonstrating exceptional generative capabilities, recent work has begun to explore training a robust diffusion model from imbalanced data. CBDM (Qin et al., 2023) employs a distribution adjustment regularizer during training to augment the minorities. Yan et al. (2024) introduce a contrastive learning regularization loss to strengthen the minorities. OC (Zhang et al., 2024) utilizes transfer learning between majorities and minorities to enhance the quality of minority generation.

## 3 PRELIMINARIES

### 3.1 PROBLEM FORMULATION

Let $\mathcal{X}$ and $\mathcal{Y} = \{1, 2, \ldots, C\}$ be the image space and the class space, where $C$ represents the class number. An imbalanced training set can be denoted as $\mathcal{D} = \{(\mathbf{x}^n, y^n)\}_{n=1}^N \in (\mathcal{X}, \mathcal{Y})^N$, where $N$ is the size of the training set. The sample number $N_c$ of each class $c \in \mathcal{Y}$ in the descending order exhibits a long-tailed distribution. The goal is to learn a generative diffusion model $p_\theta(\mathbf{x}|y)$, parameterized by $\theta$ from the imbalanced training set $\mathcal{D}$, capable of generating realistic and diverse samples across all classes. For unconditional generation using $p_\theta(\mathbf{x}|y)$, the class condition can be set to Null, resulting in $p_\theta(x) = p_\theta(x|\text{Null})$.

## 3.2 DIFFUSION MODELS

We briefly review discrete-time diffusion models, specifically denoising diffusion probabilistic models (DDPMs) (Ho et al., 2020). Given a random variable $\mathbf{x} \in \mathcal{X}$ and a *forward diffusion process* on $\mathbf{x}$ defined as $\mathbf{x}_{1:T} := \mathbf{x}_1, \ldots, \mathbf{x}_T$ with $T \in \mathbb{N}^+$, the Markov transition probability from $\mathbf{x}_{t-1}$ to $\mathbf{x}_t$ is $q(\mathbf{x}_t|\mathbf{x}_{t-1}) = \mathcal{N}(\mathbf{x}_t; \sqrt{1 - \beta_t}\mathbf{x}_{t-1}, \beta_t \mathbf{I})$, where $\mathbf{x}_0 := \mathbf{x} \sim q(\mathbf{x}_0)$, and $\{\beta_t\}_{t=1}^T$ is the variance schedule. The forward process allows us to sample $\mathbf{x}_t$ at an arbitrary timestep $t$ directly from $\mathbf{x}_0$ in a closed form $q(\mathbf{x}_t|\mathbf{x}_0) = \mathcal{N}(\mathbf{x}_t; \sqrt{\bar{\alpha}_t}\mathbf{x}_0, (1 - \bar{\alpha}_t)\mathbf{I})$, where $\alpha_t := 1 - \beta_t$ and $\bar{\alpha}_t := \prod_{i=1}^t \alpha_i$. The variance schedule is prescribed such that $\mathbf{x}_T$ is nearly an isotropic Gaussian distribution.

**Training objective.** The *reverse process* for DDPMs is defined as a Markov chain that aims to approximate $q(\mathbf{x}_0)$ by gradually denoising from the standard Gaussian distribution $p(\mathbf{x}_T) = \mathcal{N}(\mathbf{x}_T; \mathbf{0}, \mathbf{I})$: $p_\theta(\mathbf{x}_{t-1}|\mathbf{x}_t) = \mathcal{N}(p_\theta(\mathbf{x}_{t-1}; \boldsymbol{\mu}_\theta(\mathbf{x}_t, t), \sigma_t^2 \mathbf{I})$, where $\boldsymbol{\mu}_\theta(\mathbf{x}_t, t) = \frac{1}{\sqrt{\alpha_t}}(\mathbf{x}_t - \frac{\beta_t}{\sqrt{1-\bar{\alpha}_t}}\boldsymbol{\epsilon}_\theta(\mathbf{x}_t, t))$ is parameterized by a time-conditioned noise prediction network $\boldsymbol{\epsilon}_\theta(\mathbf{x}_t, t)$ and $\sigma_1, \ldots, \sigma_T$ are time dependent constants that can be predefined or analytically computed (Bao et al., 2022). The reverse process can be learned by optimizing the variational lower bound on log-likelihood as

$$\log p_\theta(\mathbf{x}) \geq \mathbb{E}_q[-D_{\mathrm{KL}}(q(\mathbf{x}_T|\mathbf{x}_0)\|p(\mathbf{x}_T)) + \log p_\theta(\mathbf{x}_0|\mathbf{x}_1) - \sum_{t>1} D_{\mathrm{KL}}(q(\mathbf{x}_{t-1}|\mathbf{x}_t, \mathbf{x}_0)\|p_\theta(\mathbf{x}_{t-1}|\mathbf{x}_t))]$$

$$= -\mathbb{E}_{\boldsymbol{\epsilon},t}[w_t\|\boldsymbol{\epsilon}_\theta(\mathbf{x}_t, t) - \boldsymbol{\epsilon}\|_2^2] + C_1, \tag{1}$$

where $\boldsymbol{\epsilon} \sim \mathcal{N}(\boldsymbol{\epsilon}; \mathbf{0}, \mathbf{1})$, $\mathbf{x}_t = \sqrt{\bar{\alpha}_t}\mathbf{x}_0 + \sqrt{1 - \bar{\alpha}_t}\boldsymbol{\epsilon}$ according to the forward process, $w_t = \frac{\beta_t^2}{2\sigma_t^2\alpha_t(1-\bar{\alpha}_t)}$, and $C_1$ is typically small and can be dropped (Ho et al., 2020; Song et al., 2021b). The term $\mathcal{L}_{\mathrm{Diff}}(\mathbf{x}, \theta) = \mathbb{E}_{\boldsymbol{\epsilon},t}[w_t\|\boldsymbol{\epsilon}_\theta(\mathbf{x}_t, t) - \boldsymbol{\epsilon}\|_2^2]$ is called the *diffusion loss* (Kingma et al., 2021). To benefit sample quality, Ho et al. (2020) apply a simplified training objective by setting $w_t = 1$.

**Class-conditional diffusion models.** When the class labels of the training set are available, the class-conditional diffusion model $p_\theta(\mathbf{x}|y)$ can be parameterized by $\boldsymbol{\epsilon}(\mathbf{x}_t, t, y)$. And the unconditional diffusion model $p_\theta(\mathbf{x})$ can be viewed as a special case with a null condition $\boldsymbol{\epsilon}(\mathbf{x}_t, t, \mathrm{Null})$. A similar lower bound on the class-conditional log-likelihood to Eq. (1) is

$$\log p_\theta(\mathbf{x}|y) \geq -\mathbb{E}_{\boldsymbol{\epsilon},t}[w_t\|\boldsymbol{\epsilon}_\theta(\mathbf{x}_t, t, y) - \boldsymbol{\epsilon}\|_2^2] + C_2, \tag{2}$$

where $C_2$ is another small constant and can be dropped (Ho et al., 2020; Song et al., 2021b). The class-conditional diffusion loss can be written as $\mathcal{L}_{\mathrm{Diff}}(\mathbf{x}, y, \theta) = \mathbb{E}_{\boldsymbol{\epsilon},t}[w_t\|\boldsymbol{\epsilon}_\theta(\mathbf{x}_t, t, y) - \boldsymbol{\epsilon}\|_2^2]$.

# 4 METHOD

## 4.1 MOTIVATION

Although diffusion models have demonstrated significant advantages in terms of fidelity and diversity in generation, most existing diffusion models implicitly assume that the training data is approximately uniformly distributed across classes. When training data exhibits real-world class imbalance, diffusion models struggle to generate high-quality and diverse samples for the minorities (Qin et al., 2023; Yan et al., 2024; Zhang et al., 2024). Current efforts focus on adjusting the objective to give more attention to minority classes, improving the robustness of diffusion models to imbalanced distributions. We tackle the robustness challenge of imbalanced distributions from the new perspective of *model capacity*. Majorities take up most of the model capacity due to quantity dominance, leaving minorities with limited capacity and poor performance. In Figure 1(a), we show the sample size for each class and the loss change after the model pruning (Han et al., 2015; Li et al., 2017) operation. It is clear that pruning has a greater impact on the output of minorities, indicating that minority classes occupy less model capacity and are therefore less robust to pruning. Jiang et al. (2021) also discuss a similar phenomenon in imbalanced discriminative models. If we can reserve and allocate a portion of the model capacity specifically for minorities, we can prevent the adverse effects of capacity domination and improve the robustness to imbalanced distributions.

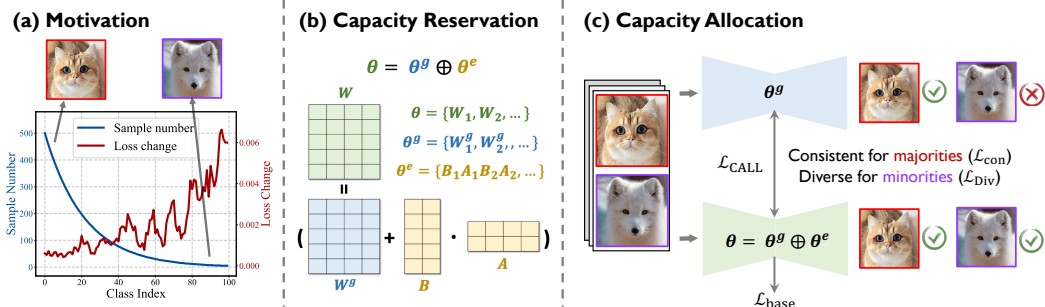

Figure 1: (a) The class distribution of training data in Imb. CIFAR-100 with imbalance ratio of $\mathrm{IR} = 100$, along with the average loss value changes per class before and after pruning the DDPM model trained on it. The x-axis shows classes arranged in descending order of sample size. The pruning rate is set to 0.1. The images are for illustration purposes only. (b) An illustration of the capacity reservation part of our method, CALL. (c) An illustration of how CALL allocates the corresponding knowledge to the reserved model capacity during training.

## 4.2 PROTECTING MINORITIES VIA CAPACITY ALLOCATION

### 4.2.1 CAPACITY RESERVATION

To allocate sufficient model capacity for minorities, we first need to explicitly partition the model capacity. Here we achieve this purpose by a technique similar to Low-Rank Adaptations (LoRAs) (Hu et al., 2022), which has demonstrated excellent performance and versatility in the field of efficient fine-tuning. While our task and goal differ, we apply its low-rank decomposition concept to partition the model capacity. For a diffusion model parameterized by $\theta = \{W_1, W_2, \ldots\}$, where each $W \in \theta$ represents a parameter matrix in the network, we decompose any $W \in \mathbb{R}^{d \times k}$ as

$$W = W^g + BA = W^g + W^e, \forall W \in \theta, \qquad (3)$$

where $W^g \in \mathbb{R}^{d \times k}$ represents the part of $W$ to be retained for majorities and generalized knowledge, $W^e = BA \in \mathbb{R}^{d \times k}$ represents the part to be allocated to the expertise of minorities, $B \in \mathbb{R}^{d \times r}$, $A \in \mathbb{R}^{r \times k}$, and the rank $r < \min(d, k)$. From Eq. (3), we decompose $\theta$ into $\theta^g = \{W_1^g, W_2^g, \ldots\}$ and $\theta^e = \{W_1^e, W_2^e, \ldots\}$ and merge them by $\theta = \theta^g \oplus \theta^e$, where $\oplus$ means the element-wise addition. An illustration of Capacity Reservation is shown in Figure 1(b).

### 4.2.2 CAPACITY ALLOCATION

With the model parameters decomposed as $\theta = \theta^g \oplus \theta^e$, our goal during training is to store minority expertise in $\theta^e$ and general knowledge in $\theta^g$, ensuring protection for minorities through capacity allocation. To achieve this, the diffusion model $p_\theta(x|y) = p_{\theta^g \oplus \theta^e}(x|y)$ should perform well on all samples, both majorities and minorities. Meanwhile, $p_{\theta^g}(x|y)$ should perform well on majorities but poorly on minorities, as $\theta^g$ is not intended to learn the minority expertise.

**Capacity allocation loss.** For $\theta = \theta^g \oplus \theta^e$, we use a loss function $\mathcal{L}_{\mathrm{base}}(\mathcal{D}, \theta)$ that balances performance across majorities and minorities. This is not our primary focus, so we directly adopt the loss functions from existing imbalanced diffusion models, *e.g.,* Zhang et al. (2024); Qin et al. (2023), as $\mathcal{L}_{\mathrm{base}}$. For imbalanced data, we propose a capacity allocation loss, which encourages $\theta^e$ to learn minority expertise and $\theta^g$ to capture general knowledge:

Capacity allocation loss: $\quad \mathcal{L}_{\mathrm{CALL}}(\mathbf{x}, y, \theta^g, \theta^e) = \mathcal{L}_{\mathrm{Con}}(\mathbf{x}, y, \theta^g, \theta^e) + \mathcal{L}_{\mathrm{Div}}(\mathbf{x}, y, \theta^g, \theta^e),$

Consistency loss: $\quad \mathcal{L}_{\mathrm{Con}}(\mathbf{x}, y, \theta^g, \theta^e) = \omega_{\mathrm{Con}}^y \mathbb{E}_t \|\boldsymbol{\epsilon}_{\theta^g \oplus \theta^e}(\mathbf{x}_t, t, y) - \boldsymbol{\epsilon}_{\theta^g}(\mathbf{x}_t, t, y)\|_2^2, \quad (4)$

Diversity loss: $\quad \mathcal{L}_{\mathrm{Div}}(\mathbf{x}, y, \theta^g, \theta^e) = -\omega_{\mathrm{Div}}^y \mathbb{E}_t \|\boldsymbol{\epsilon}_{\theta^g \oplus \theta^e}(\mathbf{x}_t, t, y) - \boldsymbol{\epsilon}_{\theta^g}(\mathbf{x}_t, t, y)\|_2^2.$

We vary the consistency class weight $\omega_{\mathrm{Con}}$ and the diversity class weight $\omega_{\mathrm{Div}}$ applied to different classes. For class $c \in \mathcal{Y}$ with $N_c$ instances, a larger $N_c$ (majorities) results in a higher consistency class weight $\omega_{\mathrm{Con}}^c$, leading to more consistent outputs between $\boldsymbol{\epsilon}_{\theta^g \oplus \theta^e}(\mathbf{x}_t, t, y)$ and $\boldsymbol{\epsilon}_{\theta^g}(\mathbf{x}_t, t, y)$. Conversely, for the diversity class weight, a smaller $N_c$ (minorities) results in a higher $\omega_{\mathrm{Div}}^c$, leading

to more diverse outputs between $\boldsymbol{\epsilon}_{\theta^g \oplus \theta^e}(\mathbf{x}_t, t, y)$ and $\boldsymbol{\epsilon}_{\theta^g}(\mathbf{x}_t, t, y)$. Thus, $p_{\theta^g}(x|y)$ excels on majorities, as its output aligns with $\theta$, but underperforms on minorities due to the divergence between the outputs of $\theta^g$ and $\theta$. Specifically,

$$\omega_{\text{Con}}^y = \frac{CN_y}{\sum_{c=1}^C N_c}, \qquad \omega_{\text{Div}}^y = \frac{C}{N_y \sum_{c=1}^C \frac{1}{N_c}}. \tag{5}$$

Here $\omega_{\text{Con}}$ scales linearly with class sample size, while $\omega_{\text{Div}}$ is inversely proportional to class sample size, ensuring $\omega_{\text{Con}} = \omega_{\text{Div}} = 1$, $\mathcal{L}_{\text{CALL}}(\mathbf{x}, y, \theta^g, \theta^e) = 0$ for a balanced training set.

**Joint optimization.** For $\theta = \theta^g \oplus \theta^e$, we optimize the base loss $\mathcal{L}_{\text{base}}$ and the capacity allocation loss $\mathcal{L}_{\text{CALL}}$, weighted by hyperparameter $\lambda$:

$$\min_\theta \mathcal{L}_{\text{Total}}(\mathcal{D}, \theta) = \mathcal{L}_{\text{base}}(\mathcal{D}, \theta) + \lambda \sum_{(\mathbf{x}, y) \in \mathcal{D}} \frac{1}{N} \mathcal{L}_{\text{CALL}}(\mathbf{x}, y, \theta^g, \theta^e), \tag{6}$$

where the base loss $\mathcal{L}_{\text{base}}$ optimizes $\theta$ for both majorities and minorities, while the capacity allocation loss $\mathcal{L}_{\text{CALL}}$ acts as a regularizer to allocate capacity and protect minorities. This guides $\theta$ toward more balanced and effective model weights. An illustration of the training process of our CALL is presented in Figure 1(c).

**Inference.** For inference, we can explicitly compute and store $\theta = \theta^g \oplus \theta^e$, and sample images from $p_\theta(\mathbf{x}|y)$. Thus, our method does not increase model capacity, ensuring *no additional inference latency* compared to a standard diffusion model, which is crucial as inference speed is a key bottleneck in real-world deployment (Song et al., 2021a). This advantage comes from using a LoRA-like parameter decomposition in Eq. (3) and explicitly aggregating the parameters during inference.

### 4.3 DISSCUSSION

**Comparison with existing imbalanced diffusion models.** Unlike current methods such as CBDM and OC, which prioritize designing more suitable objective functions for imbalanced data, our CALL improves the robustness of diffusion models to imbalanced distributions from a new perspective: allocating model capacity to protect minorities. CALL is orthogonal and can benefit from improved objective functions to achieve further enhancements (as shown empirically in Table 5).

**Comparison with LoRA.** While the capacity reservation mechanism in CALL shares a similar structure with LoRA, our goal is to decompose and allocate model capacity prior to training, whereas LoRA is aimed at efficiently fine-tuning pre-trained models. Additionally, our method involves a joint training strategy, whereas LoRA focuses solely on optimizing the low-rank components.

**Comparison with ensemble-based imbalanced classification methods.** Several ensemble-based methods (Cui et al., 2023; Wang et al., 2021b; Zhang et al., 2022) leverage multiple experts to capture diverse knowledge, achieving strong performance in classification tasks through prediction ensemble. However, most of these methods are tailored for classification networks in terms of architecture, training paradigm, and loss functions, making them unsuitable for direct application in diffusion models. While they also involve knowledge allocation, their gain mainly comes from increased capacity and ensemble predictions. Additionally, they often require structural modifications to the network and incur higher inference latency, further limiting applicability. In contrast, our method introduces no changes to network structure, does not increase model capacity or inference latency, and enhances imbalanced diffusion models purely through capacity allocation.

**Extension to LoRA-finetuning.** Our method can be seamlessly extended to LoRA-finetuning scenarios by modifying Eq. (3) to the form: $W = W^f + B^g A^g + B^e A^e$. Here, $\theta^f = \{W_1^f, W_2^f, \ldots\}$ represents the frozen pre-trained model parameters, $\theta^g = \{B_1^g A_1^g, B_2^g A_2^g, \ldots\}$ denotes the trainable parameters allocated for majorities and generalized knowledge, and $\theta^e = \{B_1^e A_1^e, B_2^e A_2^e, \ldots\}$ corresponds to the trainable parameters reserved for minority expertise. For $W \in \mathbb{R}^{d \times k}$, $B^g \in \mathbb{R}^{d \times r^g}$, $A^g \in \mathbb{R}^{r^g \times k}$, $B^e \in \mathbb{R}^{d \times r^e}$, $A^e \in \mathbb{R}^{r^e \times k}$, we have $r^e < r^g < \min(d, k)$. During inference, the model parameters are merged by $\theta = \theta^f \oplus \theta^g \oplus \theta^e$. This extension preserves the structure of LoRA while enhancing the fine-tuning process by capacity allocation for imbalanced data.

Table 1: FIDs ($\downarrow$), KIDs ($\downarrow$), Recalls ($\uparrow$), and ISs ($\uparrow$) of CALL and various baseline methods on Imb. CIFAR-10 and Imb. CIFAR-100 with different imbalance ratios IR $= \{100, 50\}$. All results are reported as Mean $\pm$ Std. **Best** and second-best results are highlighted.

| **Imb. CIFAR-10, IR $= 100$** | | | | |
|---|---|---|---|---|
| Method | FID $\downarrow$ | KID $\downarrow$ | Recall $\uparrow$ | IS $\uparrow$ |
| DDPM (Ho et al., 2020) | $10.697 \pm 0.079$ | $0.0035 \pm 0.0008$ | $0.47 \pm 0.01$ | $9.39 \pm 0.12$ |
| +ADA (Karras et al., 2020) | $9.266 \pm 0.133$ | $0.0029 \pm 0.0003$ | $0.49 \pm 0.02$ | $9.26 \pm 0.14$ |
| +RS (Mahajan et al., 2018) | $12.332 \pm 0.064$ | $0.0037 \pm 0.0003$ | $0.45 \pm 0.02$ | $9.25 \pm 0.08$ |
| +Focal (Lin et al., 2017) | $10.842 \pm 0.134$ | $0.0034 \pm 0.0001$ | $0.46 \pm 0.03$ | $9.42 \pm 0.18$ |
| CBDM (Qin et al., 2023) | $\underline{8.233 \pm 0.152}$ | $\underline{0.0026 \pm 0.0001}$ | $\mathbf{0.53 \pm 0.02}$ | $9.23 \pm 0.11$ |
| OC (Zhang et al., 2024) | $8.390 \pm 0.063$ | $0.0027 \pm 0.0002$ | $\underline{0.52 \pm 0.03}$ | $\mathbf{9.53 \pm 0.12}$ |
| CALL | $\mathbf{7.727 \pm 0.124}$ | $\mathbf{0.0023 \pm 0.0001}$ | $\mathbf{0.53 \pm 0.01}$ | $\underline{9.52 \pm 0.10}$ |
| **Imb. CIFAR-10, IR $= 50$** | | | | |
| Method | FID $\downarrow$ | KID $\downarrow$ | Recall $\uparrow$ | IS $\uparrow$ |
| DDPM (Ho et al., 2020) | $10.216 \pm 0.138$ | $0.0035 \pm 0.0002$ | $0.47 \pm 0.03$ | $9.37 \pm 0.13$ |
| +ADA (Karras et al., 2020) | $9.132 \pm 0.215$ | $0.0030 \pm 0.0002$ | $0.51 \pm 0.04$ | $9.28 \pm 0.21$ |
| +RS (Mahajan et al., 2018) | $11.231 \pm 0.177$ | $0.0038 \pm 0.0002$ | $0.47 \pm 0.02$ | $9.31 \pm 0.14$ |
| +Focal (Lin et al., 2017) | $10.315 \pm 0.263$ | $0.0034 \pm 0.0003$ | $0.48 \pm 0.01$ | $9.38 \pm 0.23$ |
| CBDM (Qin et al., 2023) | $\underline{7.933 \pm 0.082}$ | $\underline{0.0026 \pm 0.0002}$ | $\mathbf{0.54 \pm 0.02}$ | $9.42 \pm 0.14$ |
| OC (Zhang et al., 2024) | $8.034 \pm 0.225$ | $0.0027 \pm 0.0001$ | $\underline{0.53 \pm 0.01}$ | $\underline{9.65 \pm 0.09}$ |
| CALL | $\mathbf{7.372 \pm 0.125}$ | $\mathbf{0.0024 \pm 0.0002}$ | $\mathbf{0.54 \pm 0.01}$ | $\mathbf{9.69 \pm 0.09}$ |
| **Imb. CIFAR-100, IR $= 100$** | | | | |
| Method | FID $\downarrow$ | KID $\downarrow$ | Recall $\uparrow$ | IS $\uparrow$ |
| DDPM (Ho et al., 2020) | $10.163 \pm 0.077$ | $0.0029 \pm 0.0005$ | $0.46 \pm 0.01$ | $\mathbf{13.45 \pm 0.15}$ |
| +ADA (Karras et al., 2020) | $9.482 \pm 0.125$ | $0.0032 \pm 0.0002$ | $0.51 \pm 0.01$ | $12.44 \pm 0.16$ |
| +RS (Mahajan et al., 2018) | $11.432 \pm 0.287$ | $0.0038 \pm 0.0007$ | $0.44 \pm 0.03$ | $12.12 \pm 0.18$ |
| +Focal (Lin et al., 2017) | $10.212 \pm 0.110$ | $0.0032 \pm 0.0004$ | $0.47 \pm 0.02$ | $13.07 \pm 0.26$ |
| CBDM (Qin et al., 2023) | $10.051 \pm 0.391$ | $0.0036 \pm 0.0003$ | $\underline{0.51 \pm 0.01}$ | $12.35 \pm 0.12$ |
| OC (Zhang et al., 2024) | $\underline{8.309 \pm 0.233}$ | $\underline{0.0026 \pm 0.0002}$ | $\mathbf{0.52 \pm 0.02}$ | $\underline{13.44 \pm 0.20}$ |
| CALL | $\mathbf{7.519 \pm 0.132}$ | $\mathbf{0.0017 \pm 0.0003}$ | $\mathbf{0.52 \pm 0.02}$ | $\mathbf{13.45 \pm 0.23}$ |
| **Imb. CIFAR-100, IR $= 50$** | | | | |
| Method | FID $\downarrow$ | KID $\downarrow$ | Recall $\uparrow$ | IS $\uparrow$ |
| DDPM (Ho et al., 2020) | $9.363 \pm 0.069$ | $0.0032 \pm 0.0002$ | $0.47 \pm 0.02$ | $\mathbf{14.27 \pm 0.22}$ |
| +ADA (Karras et al., 2020) | $8.927 \pm 0.138$ | $0.0033 \pm 0.0001$ | $0.51 \pm 0.02$ | $12.89 \pm 0.17$ |
| +RS (Mahajan et al., 2018) | $10.259 \pm 0.217$ | $0.0037 \pm 0.0003$ | $0.47 \pm 0.03$ | $12.38 \pm 0.23$ |
| +Focal (Lin et al., 2017) | $9.477 \pm 0.114$ | $0.0034 \pm 0.0002$ | $0.49 \pm 0.03$ | $13.31 \pm 0.15$ |
| CBDM (Qin et al., 2023) | $8.946 \pm 0.178$ | $0.0036 \pm 0.0003$ | $\mathbf{0.55 \pm 0.02}$ | $12.59 \pm 0.19$ |
| OC (Zhang et al., 2024) | $\underline{7.188 \pm 0.274}$ | $\underline{0.0024 \pm 0.0002}$ | $\underline{0.54 \pm 0.01}$ | $13.99 \pm 0.22$ |
| CALL | $\mathbf{6.732 \pm 0.052}$ | $\mathbf{0.0021 \pm 0.0001}$ | $\mathbf{0.55 \pm 0.03}$ | $\underline{14.12 \pm 0.15}$ |

## 5 EXPERIMENTS

### 5.1 EXPERIMENTAL SETUP

**Datasets.** We conduct experiments on the imbalanced versions of commonly used datasets in the field of image synthesis, including CIFAR-10 (Krizhevsky et al., 2009), CIFAR-100 (Krizhevsky et al., 2009), CelebA-HQ (Karras et al., 2018), and ArtBench-10 (Liao et al., 2022). CIFAR-10 and CIFAR-100 have a resolution of $32 \times 32$, while for CelebA-HQ, we use the $64 \times 64$ version, and for ArtBench-10, we use the original resolution of $256 \times 256$. We follow Cao et al. (2019) to construct imbalanced versions of these datasets by downsampling, resulting in an exponential decrease in the sample size of each class with its index. We refer to these imbalanced datasets as Imb. dataset, *e.g.*, Imb. CIFAR-10. We control the level of imbalance in the dataset by setting different imbalance ratios IR $\in \{50, 100\}$, where IR is the ratio of the number of samples in the most populous class to that in the least populous class, defined as IR $= \frac{\max_{c \in \mathcal{Y}} N_c}{\min_{c \in \mathcal{Y}} N_c}$. For Imb. CIFAR-10 and Imb. ArtBench-10, we divide the dataset into three splits: *Many* (classes 0-2), *Medium* (classes 3-5), and *Few* (classes 6-9) based on class sizes in descending order. Similarly, for Imb. CIFAR-100, the splits are *Many* (classes 0-32), *Medium* (classes 33-65), and *Few* (classes 66-99).

Table 2: FIDs ($\downarrow$), KIDs ($\downarrow$), and per-class FIDs ($\downarrow$) of CALL and baselines on Imb. CelebA-HQ with different imbalance ratios IR $= \{100, 50\}$.

| Imb. CelebA-HQ, IR $= 100$ | | | | |
|---|---|---|---|---|
| Method | Female FID $\downarrow$ | Male FID $\downarrow$ | Overall FID $\downarrow$ | KID $\downarrow$ |
| DDPM (Ho et al., 2020) | $7.143 \pm 0.147$ | $16.425 \pm 0.032$ | $8.727 \pm 0.126$ | $0.0037 \pm 0.0001$ |
| CBDM (Qin et al., 2023) | $7.043 \pm 0.079$ | $14.273 \pm 0.183$ | $7.823 \pm 0.115$ | $0.0043 \pm 0.0002$ |
| OC (Zhang et al., 2024) | $7.092 \pm 0.323$ | $13.962 \pm 0.221$ | $7.871 \pm 0.237$ | $0.0034 \pm 0.0002$ |
| CALL | $\mathbf{6.815 \pm 0.241}$ | $\mathbf{12.788 \pm 0.316}$ | $\mathbf{7.538 \pm 0.201}$ | $\mathbf{0.0033 \pm 0.0002}$ |

| Imb. CelebA-HQ, IR $= 50$ | | | | |
|---|---|---|---|---|
| Method | Female FID $\downarrow$ | Male FID $\downarrow$ | Overall FID $\downarrow$ | KID $\downarrow$ |
| DDPM (Ho et al., 2020) | $7.348 \pm 0.219$ | $14.808 \pm 0.152$ | $8.007 \pm 0.265$ | $0.0034 \pm 0.0002$ |
| CBDM (Qin et al., 2023) | $7.317 \pm 0.273$ | $12.592 \pm 0.181$ | $7.423 \pm 0.139$ | $0.0042 \pm 0.0001$ |
| OC (Zhang et al., 2024) | $7.283 \pm 0.226$ | $12.938 \pm 0.277$ | $7.438 \pm 0.247$ | $0.0034 \pm 0.0003$ |
| CALL | $\mathbf{7.147 \pm 0.182}$ | $\mathbf{11.273 \pm 0.146}$ | $\mathbf{7.193 \pm 0.282}$ | $\mathbf{0.0033 \pm 0.0002}$ |

**Baselines.** We consider baselines including: (1) the base denoising diffusion probabilistic model (DDPM); (2) methods specifically targeting imbalanced diffusion models: the class-balancing diffusion model (CBDM) (Qin et al., 2023) and Oriented Calibration (OC) (Zhang et al., 2024); (3) applying imbalance learning methods from discriminative models or generative adversarial networks (GANs) to diffusion models: re-sampling (RS) (Mahajan et al., 2018), adaptive discriminator augmentation (ADA) (Karras et al., 2020), and focal loss (Lin et al., 2017). Note that many imbalanced learning methods for discriminative models and GANs heavily rely on specific model architectures or training paradigms, *e.g.,* Menon et al. (2021); Zhou et al. (2023); Rangwani et al. (2022), making them incompatible with imbalanced diffusion models.

**Implementation details.** Following Ho et al. (2020), we utilize a U-Net (Ronneberger et al., 2015) based on a Wide ResNet (Zagoruyko & Komodakis, 2016) as the noise prediction network. We set the hyperparameters for DDPM as $\beta_1 = 10^{-4}$ and $\beta_T = 0.02$, with maximum timestep $T = 1000$. The Adam optimizer (Kingma & Ba, 2015) is used with betas $= (0.9, 0.999)$ and a learning rate of $2 \times 10^{-4}$. The dropout rate is set to $0.1$. We use a batch size of 64 and train the model for 300,000 steps, including a warm-up period of 5,000 steps. For the rank of $BA$ in Eq. (3), we fix it at $\frac{1}{10} \min(d, k)$. We only apply the low-rank decomposition to the upsampling part of the U-Net, i.e., the latter half of the model, as the shallow layers tend to capture more general knowledge (Alzubaidi et al., 2021). For the hyperparameter $\lambda$ in Eq. (6), we fix it as $\lambda = 1$. For the base loss in Eq. (6), we adopt the objective function from Zhang et al. (2024), unless otherwise specified. During inference, new images are generated utilizing the 50-step DDIM solver (Song et al., 2021a).

**Metrics.** The performance of our method and all baselines is evaluated using the metrics Frechet Inception Distance (FID) (Heusel et al., 2017), Kernel Inception Distance (KID) (Binkowski et al., 2018), Recall (Kynkäänniemi et al., 2019), and Inception Score (IS) (Salimans et al., 2016). All metrics are calculated based on features extracted from a pre-trained Inception-V3 (Szegedy et al., 2016) model[1]. During evaluation, the metrics are calculated using a balanced set of real images and 50,000 generated images. The metrics for each {*many*, *medium*, *few*} split are computed using the corresponding split's real images and 20,000 generated images.

## 5.2 MAIN RESULTS

**Performance on Imb. CIFAR-10 and Imb. CIFAR-100.** In Table 1, we summarize the FIDs, KIDs, Recalls, ISs of our CALL and all baseline methods on Imb. CIFAR-10 and Imb. CIFAR-100 with different imbalance ratios IR $= \{50, 100\}$. Our CALL achieves the best results on 16 metrics across all four settings, except for two slightly lower ISs. Note that IS cannot detect mode collapse (Barratt & Sharma, 2018), *e.g.,* if the generated minority samples are overwhelmed by majority characteristics, such low-quality images would not lead to a drop in IS, which explains why vanilla DDPM still performs well on some ISs. Additionally, IS lacks a reference to real images, making it generally considered a less reliable metric (Borji, 2019; Nunn et al., 2021). On

---

[1]https://github.com/toshas/torch-fidelity/releases/download/v0.2.0/weights-inception-2015-12-05-6726825d.pth

Table 3: FIDs (↓), KIDs (↓), Recalls (↑), and ISs (↑) of CALL and various baseline methods on Imb. ArtBench-10 (imbalance ratios IR = {100, 50}) using LoRA to fine-tune Stable Diffusion.

| Imb. ArtBench-10, IR = 100 | | | | |
|---|---|---|---|---|
| Method | FID ↓ | KID ↓ | Recall ↑ | IS ↑ |
| DDPM (Ho et al., 2020) | 27.083 ± 0.438 | 0.0142 ± 0.0003 | 0.39 ± 0.01 | 8.47 ± 0.19 |
| CBDM (Qin et al., 2023) | 25.723 ± 0.263 | 0.0122 ± 0.0002 | 0.43 ± 0.01 | 7.97 ± 0.22 |
| OC (Zhang et al., 2024) | 24.315 ± 0.162 | 0.0106 ± 0.0005 | 0.42 ± 0.01 | **8.71 ± 0.20** |
| CALL | **22.776 ± 0.078** | **0.0087 ± 0.0002** | **00.44 ± 0.02** | **8.71 ± 0.18** |
| Imb. ArtBench-10, IR = 50 | | | | |
| Method | FID ↓ | KID ↓ | Recall ↑ | IS ↑ |
| DDPM (Ho et al., 2020) | 25.557 ± 0.082 | 0.0134 ± 0.0004 | 0.39 ± 0.02 | 8.41 ± 0.15 |
| CBDM (Qin et al., 2023) | 24.487 ± 0.153 | 0.0114 ± 0.0002 | 0.43 ± 0.02 | 8.03 ± 0.23 |
| OC (Zhang et al., 2024) | 23.287 ± 0.232 | 0.0097 ± 0.0003 | 0.43 ± 0.02 | 8.48 ± 0.17 |
| CALL | **21.733 ± 0.153** | **0.0080 ± 0.0002** | **0.44 ± 0.01** | **8.51 ± 0.23** |

the most widely used metric FID, CALL achieve significant improvements over DDPM with gains of 2.725, 2.844, 2.644, and 2.571, and consistent improvements over the best baseline in each setting by 0.506, 0.561, 0.790, and 0.456, respectively. For baseline methods, CBDM performs well on Imb. CIFAR-10, while OC shows better results on Imb. CIFAR-100. DDPM + RS generally performs worse than DDPM. DDPM + ADA, although still weaker than specialized methods like CBDM and OC, demonstrates stable improvements over DDPM, suggesting the potential of exploring data augmentation to address challenges of imbalanced data in diffusion models. DDPM + Focal achieves comparable results to DDPM, likely because the loss differences between classes in diffusion models are less distinct, making Focal loss less effective for loss-based hard example mining.

**Many/Medium/Few analysis.** In Figure 2, we show the fine-grained {*many*, *medium*, *few*} per-split FIDs of different methods on Imb. CIFAR-10 and Imb. CIFAR-100 with imbalance ratio IR = 100. Our method achieves the best results across all three splits, with the primary improvements observed in the Medium and Few classes. It is noteworthy that on Imb. CIFAR-10, the generation quality for Medium

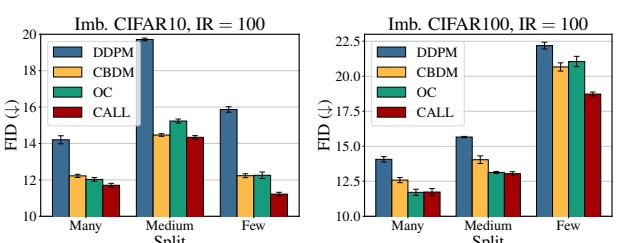

Figure 2: Per-split FIDs of CALL and baselines on Imb. CIFAR-10 (IR = 100) and Imb. CIFAR-100 (IR = 100).

classes is worse than for Few classes. Similar observations have been made on imbalanced contrastive learning (Zhou et al., 2023). This could be attributed to the inherent difficulty differences between classes, suggesting a promising direction of addressing imbalanced diffusion models by combining inherent difficulty imbalance with quantity imbalance.

**Performance on Imb. CelebA-HQ.** In Table 2, we report the FIDs, KIDs, and per-class FIDs of CALL and baseline methods on Imb. CelebA-HQ with different imbalance ratios IR = {100, 50}. Imb. CelebA-HQ contains two classes: Female and Male, with Female being the majority class. Our CALL achieves the best performance across all eight metrics in both settings. Specifically, it improves the Overall FID by 1.189 and 0.814 compared to DDPM and by 0.285 and 0.230 compared to the best baselines in each setting. For the minority class (Male), our method enhances FID by 3.637 and 3.535 over DDPM and by 1.174 and 1.319 over the best baselines. In Figure 6 in Appendix, we showcase the generated results for the "Male" class with imbalance ratio IR = 100. It is evident that our method generates more realistic and diverse faces.

**Performance of Fine-tuning Stable Diffusion on Imb. ArtBench-10.** On Imb. ArtBench-10, we fine-tune the Stable Diffusion model[2] (Rombach et al., 2022) by LoRA (Hu et al., 2022) with a rank of 128. And for $\theta^e$, the rank is set to 8. We train the model in a class-conditional manner where the textual prompt is simply set as "a {class} painting" such as "a renaissance painting". The dropout

---

[2] https://huggingface.co/lambdalabs/miniSD-diffusers

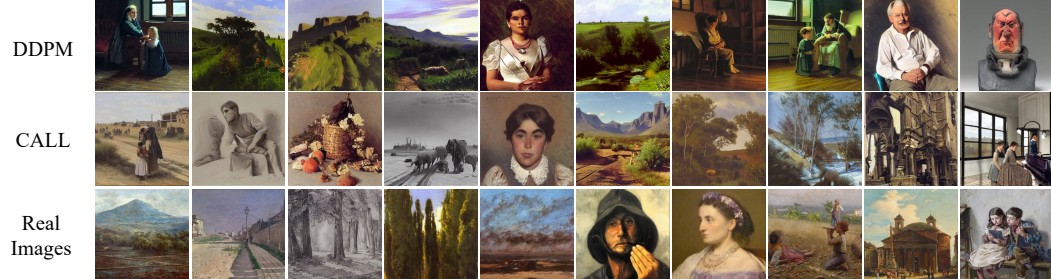

Figure 3: The visualization of generated images on Imb. ArtBench-10 with imbalance ratio IR = 100. The figure showcases the generated outputs for the class "Realism", which is one of the few classes, from both DDPM and CALL. The last row displays real images from the dataset for reference. It is evident that CALL generates results that are significantly more diverse and stylistically closer to the real images compared to DDPM. The images shown are randomly selected.

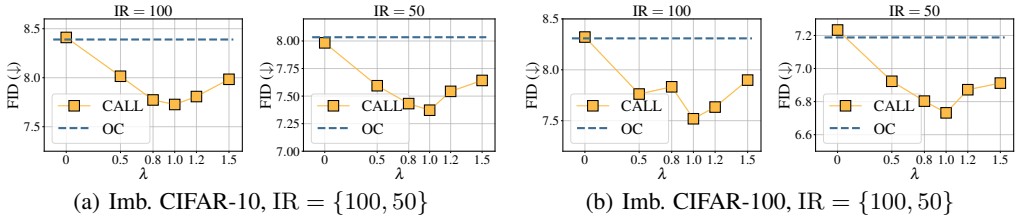

(a) Imb. CIFAR-10, IR = {100, 50}         (b) Imb. CIFAR-100, IR = {100, 50}

Figure 4: Ablation study on the hyperparameter $\lambda$ in Eq. (6). We use OC as a reference because it shows the best overall performance among the baselines and serves as our base loss. Figures (a) and (b) show results on Imb. CIFAR-10 and Imb. CIFAR-100, respectively, with imbalance ratios of IR = 100 and IR = 50 from left to right. We report FIDs for $\lambda = \{0.0, 0.5, 0.8, 1.0, 1.2, 1.5\}$.

rate is set to 0.1, and the model is trained for 100 epochs with a batch size of 64, using the AdamW optimizer (Loshchilov & Hutter, 2019) with a weight decay of $10^{-6}$ and an initial learning rate of $3 \times 10^{-4}$. During inference, we generate new images using a 50-step DDIM solver (Song et al., 2021a). In Table 3, we compare our CALL against DDPM and the two strongest baselines, CBDM and OC, on Imb. ArtBench-10 with imbalance ratios IR = {100, 50}. Our CALL achieves the best results across all eight metrics. Specifically, it outperforms DDPM in terms of FID by 4.307 and 3.824, and the best baseline in each setting by 1.539 and 1.554, respectively. Note that IS shows a decreasing trend as the imbalance ratio decreases from 100 to 50, indicating its unreliability on Imb. ArtBench. This is because the outputs of the ImageNet-pretrained Inception-V3 are less reliable for artwork images, and IS does not use real images as a reference. The generated images for one of the few classes "Realism" on Imb. ArtBench-10 with IR = 100 are shown in Figure 3. Our method generates more diverse images, and the generated styles are closer to the real images.

## 5.3 FURTHER ANALYSIS

**CALL as a universal framework.** Table 5 summarizes the performance of our CALL when integrated with DDPM, CBDM, and OC (*i.e.,* using the corresponding objective function for $\mathcal{L}_{\text{base}}$ in Eq. (6)) on Imb. CIFAR-100 with IR = 100. It can be observed that our method consistently improves the performance of imbalanced generation when combined with different baselines. Due to the orthogonality of CALL to existing methods, it can consistently benefit from improved objective functions, including potential future advancements.

**Effect of knowledge allocation between $\theta^g$ and $\theta^e$.** To investigate the effect of CALL on knowledge allocation between $\theta^g$ and $\theta^e$, we present the results of generating images using only $\theta^g$ (CALL ($\theta^g$)) and using $\theta = \theta^g \oplus \theta^e$ (CALL) on Imb. CIFAR-100 with imbalance ratio IR = 100 in Table 4. CALL ($\theta^g$) performs well on the Many and Medium classes but struggles with the few classes. In contrast, CALL shows strong performance across all splits. This indicates that CALL successfully allocates minority expertise to $\theta^e$, while reserving majority and general knowledge for $\theta^g$.

Table 4: Per-split FIDs and overall FIDs ($\downarrow$, Mean$\pm$Std) of DDPM, CALL ($\theta^g$), and CALL on Imb. CIFAR-100 with imbalance ratio IR $= 100$. Many, Medium, and Few are the three splits based on the training imbalance. **Best** results are highlighted.

| Method | Many FID $\downarrow$ | Med. FID $\downarrow$ | Few FID $\downarrow$ | Overall FID $\downarrow$ |
|---|---|---|---|---|
| DDPM (Ho et al., 2020) | $14.068 \pm 0.193$ | $15.660 \pm 0.047$ | $22.188 \pm 0.241$ | $10.163 \pm 0.077$ |
| CALL ($\theta^g$) | $11.923 \pm 0.139$ | $14.872 \pm 0.157$ | $29.357 \pm 0.318$ | $13.732 \pm 0.240$ |
| CALL ($\theta = \theta^g \oplus \theta^e$) | $\mathbf{11.732 \pm 0.247}$ | $\mathbf{13.043 \pm 0.138}$ | $\mathbf{18.729 \pm 0.141}$ | $\mathbf{7.519 \pm 0.132}$ |

Table 5: FIDs ($\downarrow$), KIDs ($\downarrow$), Recalls ($\uparrow$), and ISs ($\uparrow$) of different baselines on Imb. CIFAR-100 with imbalance ratio IR $= 100$ and their results when combined with CALL. The last two rows show the results of CALL after removing $\mathcal{L}_{\mathrm{Con}}$ and $\mathcal{L}_{\mathrm{Div}}$, respectively.

| Method | FID $\downarrow$ | KID $\downarrow$ | Recall $\uparrow$ | IS $\uparrow$ |
|---|---|---|---|---|
| DDPM (Ho et al., 2020) | $10.163 \pm 0.077$ | $0.0029 \pm 0.0005$ | $0.46 \pm 0.01$ | $\mathbf{13.45 \pm 0.15}$ |
| + CALL | $9.281 \pm 0.251$ | $0.0027 \pm 0.0002$ | $0.49 \pm 0.01$ | $13.37 \pm 0.19$ |
| CBDM (Qin et al., 2023) | $10.051 \pm 0.391$ | $0.0036 \pm 0.0003$ | $0.51 \pm 0.01$ | $12.35 \pm 0.12$ |
| + CALL | $8.837 \pm 0.245$ | $0.0029 \pm 0.0001$ | $\underline{0.51 \pm 0.02}$ | $13.07 \pm 0.16$ |
| OC (Zhang et al., 2024) | $8.309 \pm 0.233$ | $0.0026 \pm 0.0002$ | $\mathbf{0.52 \pm 0.02}$ | $\underline{13.44 \pm 0.20}$ |
| + CALL | $\mathbf{7.519 \pm 0.132}$ | $\mathbf{0.0017 \pm 0.0003}$ | $\mathbf{0.52 \pm 0.02}$ | $\mathbf{13.45 \pm 0.23}$ |
| + CALL w/o $\mathcal{L}_{\mathrm{Con}}$ | $8.412 \pm 0.227$ | $0.0029 \pm 0.0002$ | $0.50 \pm 0.01$ | $13.23 \pm 0.22$ |
| + CALL w/o $\mathcal{L}_{\mathrm{Div}}$ | $\underline{8.073 \pm 0.174}$ | $\underline{0.0025 \pm 0.0001}$ | $0.51 \pm 0.01$ | $13.42 \pm 0.16$ |

**Ablation on the hyperparameter $\lambda$ in Eq. (6).** To investigate the impact of the hyperparameter $\lambda$, the weight of the CALL loss in Eq. (6), on the performance of our method, we conduct ablation experiments on Imb. CIFAR-10 and Imb. CIFAR-100 with different imbalance ratios IR $= \{100, 50\}$. Figure 4 illustrates how the FID of CALL changes with varying $\lambda$ values under different settings. We observe that CALL maintains a consistent advantage over OC across a wide range of $\lambda$ values, with its performance peaking around $\lambda = 1.0$.

**Ablation on $\mathcal{L}_{\mathrm{Con}}$ and $\mathcal{L}_{\mathrm{Div}}$.** Table 5 presents the results of CALL as well as the ablation study where the consistency loss $\mathcal{L}_{\mathrm{Con}}$ and the diversity loss $\mathcal{L}_{\mathrm{Div}}$ are removed separately from CALL. Since $\mathcal{L}_{\mathrm{Con}}$ and $\mathcal{L}_{\mathrm{Div}}$ are responsible for allocating majority knowledge and minority expertise, respectively, removing either leads to a significant drop in performance, highlighting their necessity.

**Ablation on network configurations.** We conduct experiments on UNet architectures with varying widths and depths. Figure 5 shows FIDs on Imb. CIFAR-10 and Imb. CIFAR-100 with IR $= 100$. Different widths and depths are achieved by setting the channel_multipliers parameter to $[1, 2, 2]$, $[1, 2, 2, 2]$ (default), and $[1, 2, 3, 4]$. As shown, our method consistently demonstrates clear advantages across different network configurations.

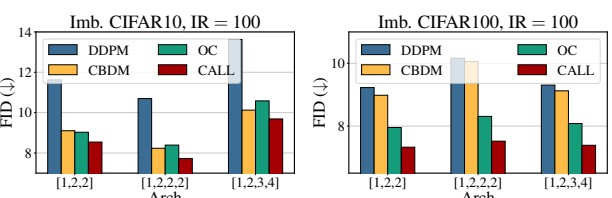

Figure 5: FIDs with various UNet configurations on Imb. CIFAR-10 and Imb. CIFAR-100 with IR $= 100$.

## 6 CONCLUSION

In this paper, we seek to improve the robustness of diffusion models to imbalanced data. Unlike previous work that focuses on improving objective functions, we aim to protect the generation performance of minorities by reserving and allocating model capacity for them. We first decompose the model parameters into parts that capture general and majority knowledge, and a dedicated part for minority expertise using low-rank decomposition techniques. By introducing a capacity allocation loss, we successfully allocate the corresponding knowledge to the reserved model capacity during training. Extensive experiments and empirical analyses confirm that our method CALL effectively protects minorities in imbalanced diffusion models via capacity allocation.

ETHICS STATEMENT

In this paper, we propose a method to enhance the robustness of generative diffusion models against imbalanced data distributions. This advancement holds significant social implications, both positive and negative. On the positive side, our approach could democratize access to high-quality data generation, allowing marginalized communities to benefit from more equitable representation in AI applications. By improving the model's performance on underrepresented classes, we can foster inclusivity in various fields, such as healthcare, finance, and education, where data-driven decisions can impact lives. Conversely, there are potential negative consequences to consider. As generative models become more powerful, they may be misused to create deceptive content, leading to misinformation and erosion of trust in digital media. Additionally, our method's emphasis on underrepresented segments in the training data poses a risk of data poisoning if supervision is lacking. Malicious actors could exploit this focus to introduce biased or harmful data, compromising the model's integrity. This vulnerability underscores the need for robust monitoring and validation mechanisms to ensure data reliability, as any compromise could lead to unintended negative consequences. Therefore, proactive data governance is essential to mitigate these risks while maximizing the benefits of our proposed method.

REPRODUCIBILITY STATEMENT

To ensure the reproducibility of experimental results, we will provide a link for an anonymous repository about the source codes of this paper in the discussion forum according to the ICLR 2025 Author Guide. All the experiments are conducted on NVIDIA A100s with Python 3.8 and Pytorch 2.0.1. We provide experimental setups and implementation details in Section 5.1 and Section 5.2.

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

---

**Algorithm 1** Algorithm of CALL

> ▷ Training, take DDPM as base, sample-wise

**Initialize**: $\theta^g = \{W_1^g, W_2^g, \ldots\}$, $\theta^e = \{B_1^e A_1^e, B_2^e A_2^e, \ldots\}$
**repeat**
  Sample data $(\mathbf{x}, y) \in \mathcal{D}$
  Sample a timestep $t \sim \mathrm{Uniform}(\{1, \ldots, T\})$
  Sample a noise $\boldsymbol{\epsilon} \sim \mathcal{N}(\mathbf{0}, \mathbf{I})$
  Base loss: $\mathcal{L}_{\mathrm{base}} = \|\boldsymbol{\epsilon}_{\theta^g \oplus \theta^e}(\sqrt{\bar{\alpha}_t}\mathbf{x} + (1 - \bar{\alpha}_t)\boldsymbol{\epsilon}, t, y) - \boldsymbol{\epsilon}\|_2^2$
  Capacity allocation loss: $\mathcal{L}_{\mathrm{CALL}} = (\omega_{\mathrm{Con}}^y - \omega_{\mathrm{Div}}^y)\|\boldsymbol{\epsilon}_{\theta^g \oplus \theta^e}(\mathbf{x}_t, t, y) - \boldsymbol{\epsilon}_{\theta^g}(\mathbf{x}_t, t, y)\|_2^2$.
  Take gradient descent on $\nabla_{\theta^g, \theta^e}(\mathcal{L}_{\mathrm{base}} + \lambda \mathcal{L}_{\mathrm{CALL}})$
**until** converged

> ▷ Sampling, take DDPM for example, sample-wise

Merge model parameters as $\theta = \theta^g \oplus \theta^e$
Sample $\mathbf{x}_T \sim \mathcal{N}(\mathbf{0}, \mathbf{I})$
**for** $t = T, \ldots, 1$ **do**
  $\mathbf{z} \sim \mathcal{N}(\mathbf{0}, \mathbf{I})$ if $t > 1$, else $z = \mathbf{0}$
  $\mathbf{x}_{t-1} = \frac{1}{\sqrt{\alpha_t}}(\mathbf{x}_t - \frac{\beta_t}{\sqrt{1-\bar{\alpha}_t}}\boldsymbol{\epsilon}_\theta(\mathbf{x}_t, t, y)) + \sigma_t \mathbf{z}$
**end for**
**return** $\mathbf{x}_0$

---

DDPM

CALL

Real Images

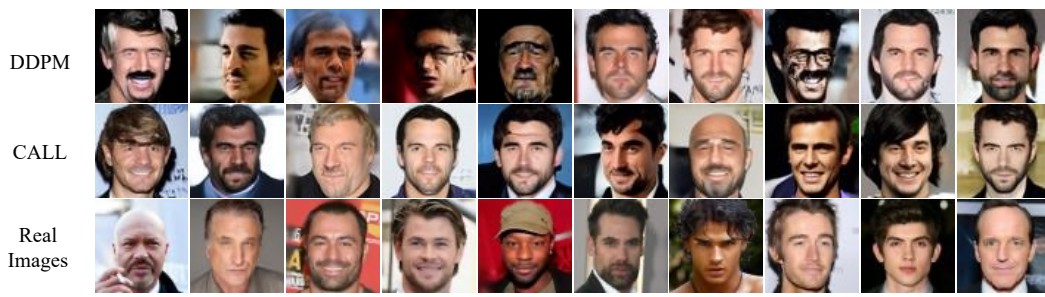

Figure 6: The visualization of generated images on Imb. CelebA-HQ with imbalance ratio IR $=$ 100. The figure showcases the generated outputs for the class "Male", which is the minority class, from both DDPM and CALL. The last row displays real images from the dataset for reference. It is evident that CALL generates generates more realistic and diverse faces.

## A  ALGORITHM

We summarize the procedure of our CALL in Algorithm 1, where we use DDPM as the base loss, employ DDPM for sampling, and illustrate the process in a sample-wise manner as an example.

## B  MORE VISUALIZATION

The generated images for one the medium classes "surrealism" on Imb. ArtBench-10 with IR $= 100$ are shown in Figure 7. It is evident that the generated styles of CALL are much closer to the real images. More visualization of generation results with CALL are presented in Figure 8 (Imb. CIFAR-100, IR $= 100$), Figure 9 (Imb. CelebA-HQ, IR $= 100$), and Figure 10 (Imb. ArtBench-10, IR $= 100$).

DDPM
CALL
Real
Images

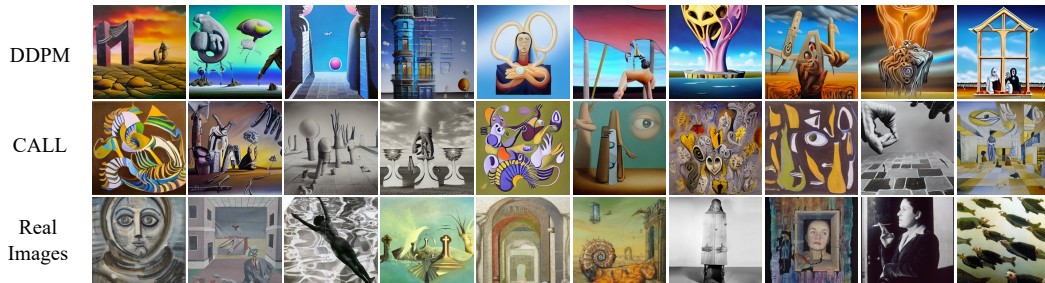

Figure 7: The visualization of generated images on Imb. ArtBench-10 with imbalance ratio IR = 100. The figure showcases the generated outputs for the class "Surrealism", which is one of the medium classes, from both DDPM and CALL. The last row displays real images from the dataset for reference. It is evident that CALL generates results that are significantly more stylistically closer to the real images compared to DDPM. The images shown are randomly selected.

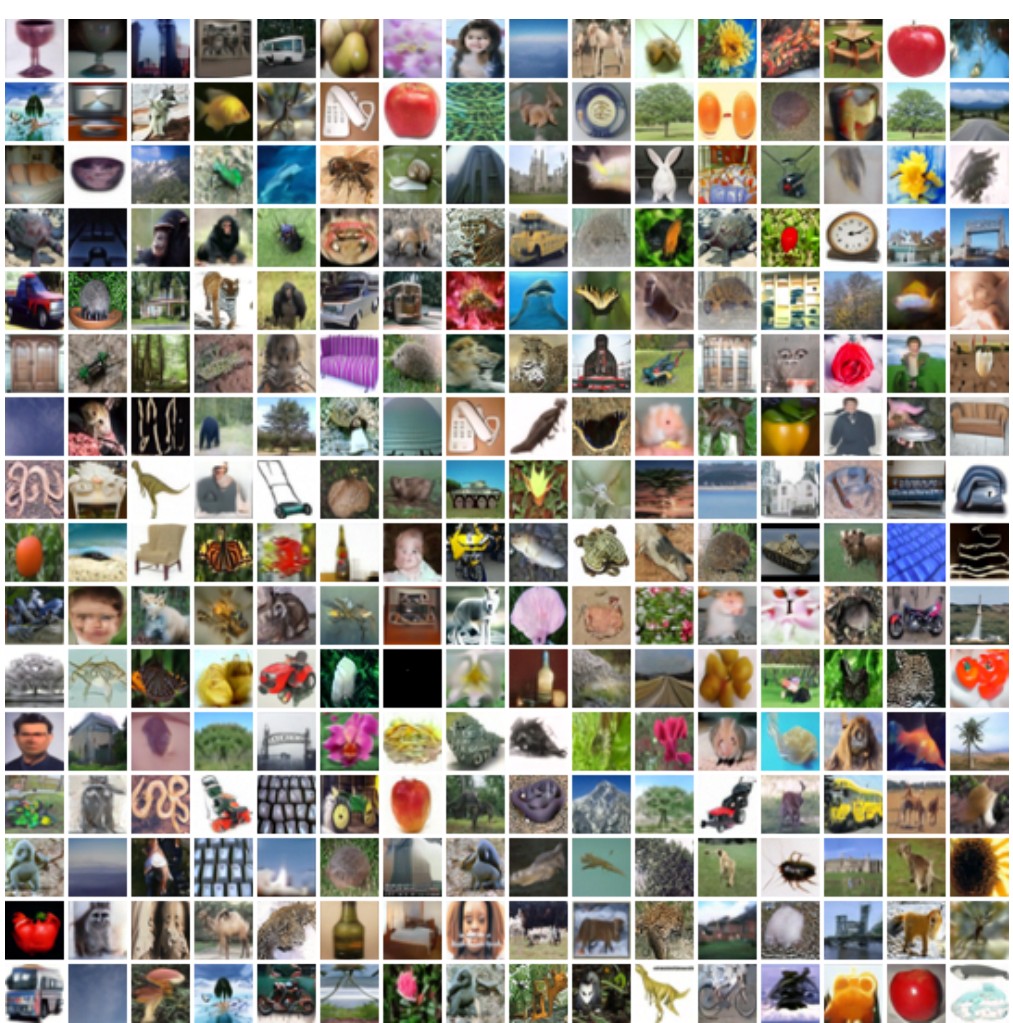

Figure 8: Visualization of generation results on Imb. CIFAR-100 (IR = 100) with CALL.

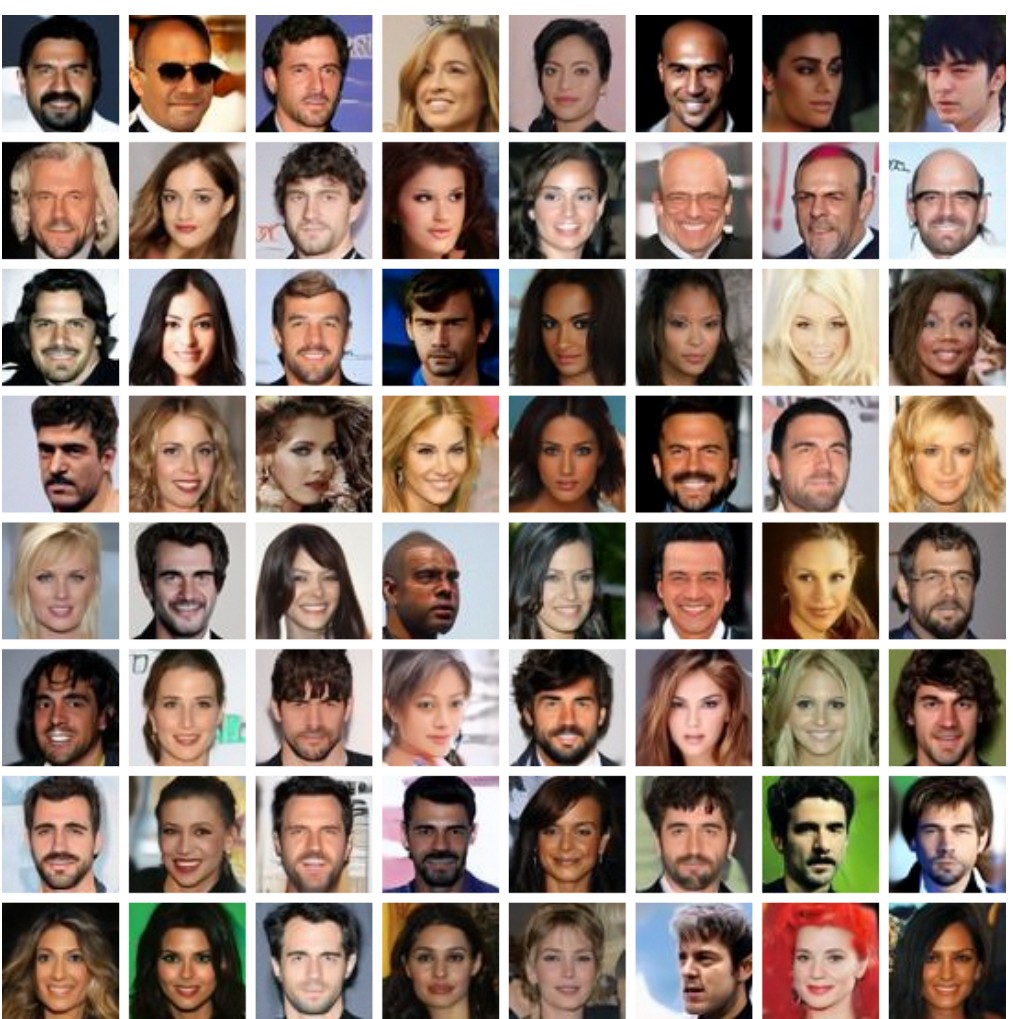

Figure 9: Visualization of generation results on Imb. CelebA-HQ (IR = 100) with CALL.

918
919
920
921
922
923
924
925
926
927
928
929
930
931
932
933
934
935
936
937
938
939
940
941
942
943
944
945
946
947
948
949
950
951
952
953
954
955
956
957
958
959
960
961
962
963
964
965
966
967
968
969
970
971

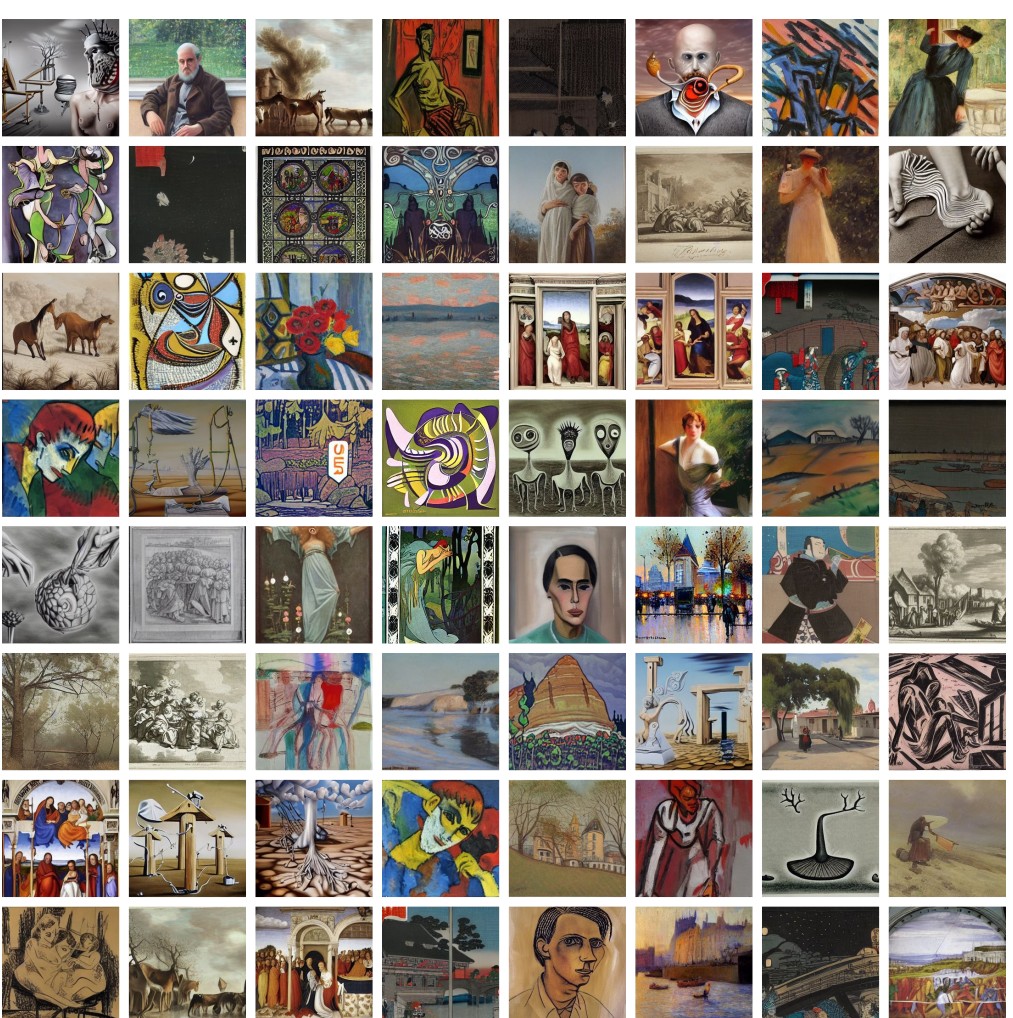

Figure 10: Visualization of generation results on Imb. ArtBench-10 (IR = 100) with CALL.

