# OpenReview forum: "Protecting Minorities in Diffusion Models via Capacity Allocation"
_ICLR.cc/2025/Conference — ICLR 2025 Conference Withdrawn Submission_

### Official Review · Reviewer_AiBf · 2024-10-26

**Soundness:** 2
**Presentation:** 3
**Contribution:** 3
**Rating:** 5
**Confidence:** 3

**Summary:**

This paper mainly aims to address the class imbalance problem in diffusion models. Although existing approaches address this issue by designing objective functions, the proposed approach handles it from the perspective of model capacity about minority classes. The authors observed that the model tends to use low capacity for minority classes and alleviate this problem by introducing additional parameters for minority classes like LoRA. Specifically, the proposed method includes introducing extra parameters that focus on learning the minority classes while devising losses that encourage the original parameters to learn more from the major classes. This ensures that the additional parameters are primarily utilized for learning the minority classes. In the experiment, the authors show the effectiveness of the proposed method across various datasets.

**Strengths:**

* The proposed method is novel in that it addresses the class imbalance in diffusion models from a capacity perspective, which significantly differs from existing methods.
* The method is straightforward and involves a small amount of additional cost, making it practical.
* The authors show the effectiveness of the proposed method across various benchmark datasets, and the method can improve the performance further by combining it with other methods.

**Weaknesses:**

* There is insufficient explanation or supporting evidence for the key hypotheses, where the model has insufficient capacity to learn minority classes due to the class imbalance, and no experiments demonstrate that the proposed method effectively addresses this limitation. First, in Figure 1 (a), the authors do not provide the experimental details such as the pruning method and experimental setup. Additionally, observing the loss change before and after pruning as an indicator of insufficient capacity appears too implicit. A more explicit experiment demonstrating the model's capacity limitations would strengthen this claim. Finally, as the proposed method is designed to alleviate these capacity issues, an experiment should be provided to show that minority classes utilize more capacity when the method is used. Note that, for a fair comparison,  the overall model capacity before and after using the proposed method should be kept consistent.
* The proposed method introduces additional parameters ($\theta^e$), which can contribute to performance improvements. However, these gains have not been considered in the comparison. To ensure a fair comparison, the baselines such as “OC + additional parameters” should also be compared with the suggested method.
* The ablation study for consistency and diversity losses is only shown in OC, while the main results use DDPM, which seems somewhat inconsistent. It might be more natural to show the ablation study on consistency and diversity losses directly in DDPM.
* Improving readability in Figure 3 and the qualitative comparison would be helpful. Indicating whether each class is major or minor and class names would make it easier to verify if the baseline models generate minor classes similarly to major classes.

**Questions:**

* An ablation study on rank could be beneficial for practitioners. Understanding how robust performance remains as rank decreases would be valuable, as practitioners often aim to use the smallest possible rank.
* In real-world scenarios, imbalance ratios are often more extreme. Demonstrating strong performance in such cases could further enahnce the effectiveness of the proposed method. Given that CIFAR-10 has ample data per class, it would be interesting to show results for imbalance ratios of 500 and 1000 as well.

---

### Official Review · Reviewer_2UPn · 2024-10-27

**Soundness:** 2
**Presentation:** 3
**Contribution:** 2
**Rating:** 5
**Confidence:** 4

**Summary:**

To address the problem of diffusion models' performance degradation on minority data due to imbalanced training data, the authors approach the issue from the perspective of model capacity. They propose the CALL loss, which implements a low-rank decomposition of the diffusion model parameters to emphasize knowledge of minority classes.

**Strengths:**

1. This work addresses an important problem.
2. The paper's perspective is novel although the method itself (LoRA) is not new. The author's experiments are comprehensive, including necessary experiments on parameters, architecture, and ablation studies.
3. The paper is clearly written.

**Weaknesses:**

1. For the motivation,

(a) The conclusion drawn from Figure 1(a), suggesting that minority classes utilize less of the model's capacity due to larger loss changes after pruning, lacks rigor. This observation doesn't rule out alternative explanations, such as minority classes potentially requiring more model parameters due to complex features, thus being more sensitive to parameter perturbations (pruning). To establish a strong link between sample size and model capacity, it would be better to control for inputs while varying only sample sizes. For instance, comparing model capacity differences when the same class is treated as a majority or minority class may be more convincing.

(b) The authors claim that *most current research on imbalanced learning cannot be directly applied to diffusion models due to structural and process differences*. However, in the field of discriminative model robustness, some methods attempt to address imbalanced training data distributions, such as apply Mixup [1] strategy for training data. It's unclear why these methods couldn't be adapted for diffusion models. More directly, would applying a simple method like Mixup to balance the training data distribution (given the class) mitigate the performance degradation in diffusion models?

(c) While I agree that model capacity offers a novel perspective, I find myself seeking further clarification on the necessity of CALL, especially given that CALL's performance is not significantly superior to existing baselines like CBDM and OC (Tables 1 and 2). It would be better if the authors could provide additional insights or evidence to further show the necessity of CALL compared to existing methods.

2. For the method,

(a)  The CALL loss may result in relatively large classes being considered as minority classes , which may not always be ideal. For example, by equations (4) and (5), we know that the loss weight of CALL is:

$w^y_{con}-w^y_{Div}  \propto N^2_y \sum_c \frac{1}{N_c} - \sum_c N_c$.

According to the authors, when  $w^y_{con}-w^y_{Div} \leq 0$, CALL loss encourages diverse outputs for minority data; otherwise, it encourages more consistent outputs for majority data.

By Jensen inequality,

$N_y \leq \sqrt{\frac{ \sum_c N_c}{\sum_c \frac{1}{N_c}}} \leq \mathbb{E}[N_c]$

This suggests that classes with sample sizes below the mean may be considered as minority classes. For example, consider a dataset with 8 classes, where the sample sizes are 25500, 24800, 13000, 12000, 11000, 8000, 4000, and 1700, respectively. The mean is 12500, which means that classes with sample sizes of 13000 and 12000 might be classified as different majority and minority classes, respectively, despite their sample sizes differing by only 1000. Therefore, the CALL method should consider more complex criteria when defining minority classes, rather than relying solely on a simple division based on the mean sample size.

(b) What is the intuition behind the weights defined in equation (5) for the CALL loss? For instance, why is $\frac{1}{N_c}$ considered?

(c)  Regarding loss (4), is it possible to consider designing the loss from the perspective of $\theta_e$? For example, consider using $\mathbb{E}||\epsilon_{\theta^g,\theta^e}-\epsilon_{\theta^e}||^2$

3.  For experiments,

(a) I notice that Figure 2 does not show loss for the majority group. Intuitively, if the majority group sacrifices some capacity, should it lead to a degradation in performance (please correct me if I'm wrong)？ Similar phenomena, where improvements in minority groups come at the cost of decline in majority groups, have been observed in other works [2,3,4]. Can the authors provide some explanation for this?

(b) Minor issues:

- What is the basis for dividing the dataset into three splits in the experiments (Line 322 and 323)?
- I appreciate the section CALL as a universal framework. The paper partially demonstrates the integration of CALL with DDPM, CBDM, and OC. If the authors could provide the detailed loss functions for these three methods in the main text, it can help readers better understand the orthogonality of CALL.

[1] Yao H, Wang Y, Li S, et al. Improving out-of-distribution robustness via selective augmentation[C]//International Conference on Machine Learning. PMLR, 2022: 25407-25437.

[2] Liu, Evan Z., et al. "Just train twice: Improving group robustness without training group information." *International Conference on Machine Learning*. PMLR, 2021.

[3] Sagawa, Shiori, et al. "Distributionally Robust Neural Networks." *International Conference on Learning Representations*.

[4] Michael Zhang, Nimit S Sohoni, Hongyang R Zhang, Chelsea Finn, and Christopher Ré. Correct-n-contrast: A contrastive approach for improving robustness to spurious correlations. In ICML, 2022.

**Questions:**

See my Weaknesses

---

### Official Review · Reviewer_HFyJ · 2024-10-28

**Soundness:** 2
**Presentation:** 3
**Contribution:** 2
**Rating:** 3
**Confidence:** 4

**Summary:**

This paper explores a class imbalance challenge in image generation using diffusion models. The problem is that generated samples of minority classes containing limited numbers of instances in the training data often exhibit degraded quality and diversity. The authors argue that the issue is due to biased capacity allocations against minority groups, where majority group members take dominant parts of the model capability. To address this, they perform low-rank decomposition of the parameters of diffusion models and develop a training loss to encourage an assignment of the detached low-rank weights exclusively to minority group members. The experimental sections of the paper exhibit the effectiveness of the proposed framework compared to existing techniques including ones tailored for the class imbalanced problem.

**Strengths:**

1. To the reviewer’s best knowledge, the authors take a novel perspective (i.e., capacity allocation) to counter the class imbalance issue in generative models.
2. The idea of incorporating low-lank decomposition exclusively for minority classes is interesting.
3. The presentation of the paper is clear and easy-to-follow.

**Weaknesses:**

1. The reviewer is not fully convinced with the authors’ key claim that poor minority generation comes from the unbalanced capacity. Fig. 1(a) is the sole evidence presented to support this assertion, yet it lacks sufficient clarity to effectively substantiate the claim. Validation would be strengthened, for instance, by extending the analysis in Fig. 1(a) to various pruning scenarios, showing consistent results across diverse pruning ratios and techniques that minorities are indeed more vulnerable than majorities.
2. The paper lacks theoretical background. Unlike prior arts tackling the same issue (e.g., CBDM [1]), the proposed training techniques are built with heuristic intuitions (which is unclear in the reviewer’s viewpoint, as mentioned in W1), thereby raising concerns on the robustness and generalizability in other unexplored benchmarks.
3. Empirical benefits compared to the prior arts [1,2] is unclear. Specifically, the reviewer concerns that whether the prominent performance values reported in the main experimental sections (e.g., Tab. 1, 2 and 3) are mostly contributed by the use of the prior SOTA (i.e., OC) as the base loss. This is hinted in the FID values in Table 5, where OC improves DDPM from 10.163 to 8.309, whereas CALL falls behind OC with the enhancement to 9.281.
4. The reviewer concerns that the paper has a fairness issue in comparison, specifically with the previous works. As pointed in W3, the key performance results (exhibited as outperforming the baselines) are due to the integration of the baseline techniques (e.g., OC [2]) with the proposed training loss. In the same perspective, the authors should also explore combinations of prior techniques, which may be orthogonal to each other and therefore can yield performance advantages.
5. For other comments, the numbers reported in this paper are not consistent in the ones appeared in the prior works [1,2]. For instance in Tab. 1, the FID value of CBDM on CIFAR-100-LT (IR=100) is 10.051, whereas it is reported as 7.82 in [2] (see Tab. 2 therein). Also, Section 4.1 seems redundant, since the same point is already covered in the introduction.
---
[1] Class-Balancing Diffusion Models, CVPR 2023

[2] Long-tailed Diffusion Models with Oriented Calibration, ICLR 2024

**Questions:**

See the weaknesses above.

---

### Official Review · Reviewer_R8SM · 2024-11-02

**Soundness:** 2
**Presentation:** 3
**Contribution:** 1
**Rating:** 5
**Confidence:** 3

**Summary:**

The paper addresses the challenge of training/fine-tuning-data imbalance for Diffusion-based image generation models. Current research targets modifications to the loss function. As an alternative, the authors explore the role of the model’s representation capacity, which the authors claim is occupied primarily by the majority classes. They empirically show this by pruning the model and testing the generation performance on minority classes (Fig. 1(a)). Minority classes experience a disproportionate increase in loss, as their cardinality in the training set reduces. To tackle this issue, the authors propose decomposing the weight matrices to explicitly “allocate” model capacity to minority classes. They further augment the loss function by introducing a consistency and a diversity term, based on the arithmetic and harmonic means of the class frequencies. The decomposition is extended to the setting of LoRA fine-tuning and is integrated with pre-existing generation models.

**Strengths:**

S1 The paper is nicely structured and easy to follow.

S2 Experimental details are present in the paper.

S3 Experiments have been conducted on diverse datasets.

S4 The results show a noticeable improvement in several cases over the chosen baselines (DDPM, CBDM, OC).

S5 The extension to LoRA FT is practical.

S6 Since the existing weight matrices are factorized, there is no increase in inference latency.

S7 The methodology can be used to extend existing architectures.

**Weaknesses:**

- W1 The novelty of this work is unclear, as it heavily relies on a distributed loss function for capacity allocation, raising questions about the "orthogonality" claim (despite improvements in models retrofitted with CALL). Additionally, the rank decomposition of weight matrices for training is not new.
- W2 The Imbalance Ratio (IR) may be an inadequate metric for assessing CALL and existing methods, as it only considers the most and least populous classes, and the chosen IR values lack clear justification. Experiments should instead fix a sample budget and vary class frequencies (or some measure of "tailedness") for a better evaluation.
- W3 Comparisons against approaches that modify the denoising or sampling process, e.g., Sehwag et al. [1] (not cited) and Yan et al. [2] (cited), should be included.
- W4 Beyond showing qualitative generation results, the impact on downstream tasks, e.g., on medical imaging datasets, should be added to round out the practical benefits.

[1] Sehwag et al.; Generating High Fidelity Data from Low-density Regions using Diffusion Models (2022); Available: https://arxiv.org/abs/2203.17260

[2] Yan et al.; Training Class-Imbalanced Diffusion Model Via Overlap Optimization (2024); Available: https://arxiv.org/abs/2402.10821

**Questions:**

- Q1 In Fig. 1(a), the change in loss displays an oscillatory behavior for classes with close to half the number of samples of the most populous class. Can the authors comment on why this is the case? Does it only happen for CIFAR-100?
- Q2 How does the rank, $r$, affect the representation capacity of the model?
- Q3 What do the authors mean by “generalized knowledge” on Line 191 (Page 4)?
- Q4 What do the authors mean by “inherent difficulty differences” on Line 415 (Page 8)?
- Q5 How were the pictures in Figures 8, 9 & 10 selected?
- Q6 Could the authors analyze the SVD of the output weight matrices ($g$, $e$, $g + e$) to better substantiate inter-class separability of their decomposition, particularly in the "Few" class scenario for Imbalanced CIFAR-10, where this analysis could be easily conducted?

- Please include the expression for $\mathscr{L}_{\text{base}}$ in the paper.
- On Line 790 on Page 15, “generates” appears twice.

---

### Note · Authors · 2024-11-15

I have read and agree with the venue's withdrawal policy on behalf of myself and my co-authors.